



# A framework for three-dimensional dynamic modeling of mountain glaciers in the Community Ice Sheet Model (CISM v2.2)

Samar Minallah[1,*], William H. Lipscomb[1,*], Gunter Leguy[1], and Harry Zekollari[2,3]

[1]Climate and Global Dynamics Laboratory, NSF National Center for Atmospheric Research, Boulder, CO, USA
[2]Department of Water and Climate, Vrije Universiteit Brussel, Brussels, Belgium
[3]Laboratory of Hydraulics, Hydrology and Glaciology (VAW), ETH Zürich, Zurich, Switzerland
[*]These authors contributed equally to this work.

**Correspondence:** Samar Minallah (minallah@ucar.edu)

**Abstract.** It is essential to improve our understanding of mountain glaciers and their effects on sea level, ecosystems, and freshwater resources in a changing climate. To this end, we implemented a framework for three-dimensional, high-resolution, regional-scale glacier simulations in the Community Ice Sheet Model (CISM v2.2), using higher-order ice-flow dynamics previously applied to the Greenland and Antarctic ice sheets. Here, we present the modeling framework and its application to the

European Alps glaciers at a 100-meter resolution, using protocols from the third phase of the Glacier Model Intercomparison Project (GlacierMIP3). The model results align well with observations and other glacier models, showing that Alpine glaciers will lose nearly half of their present-day area and volume under current climate conditions, with a near-total ice loss expected in warmer scenarios. This new development integrates glacier and ice sheet systems in a common modeling framework and will support advances in coupled land ice – Earth system assessments across timescales in the Community Earth System Model

(CESM).

## 1 Introduction

Numerical models are powerful tools to study the physical processes governing glaciers and continental ice sheets, enabling the prediction of cryospheric changes from seasonal to millennial timescales and their impacts on Earth systems. These impacts

span local, regional, and global scales and influence sea levels, hydroclimates, ecosystems, and human activities. For example, the Greenland and Antarctic ice sheets regulate global-scale climate and ocean circulation patterns and are major contributors to sea-level rise (Fox-Kemper et al., 2021). High-latitude (polar and sub-polar) glaciers, decoupled from the ice sheets, are also large contributors to sea-level rise and freshwater flux to the oceans (Hock et al., 2019; Marzeion et al., 2020; Zemp et al., 2019). Inland mountain glaciers contribute less to sea level but are important freshwater sources for streams, rivers, and lakes.

These mountain glaciers regulate ecosystems, contribute to local and downstream water supply, and interact with regional hydroclimates on sub-seasonal to decadal timescales (Huss and Hock, 2018; Milner et al., 2009; Hock et al., 2005; Ficetola



et al., 2024; Immerzeel et al., 2020; Bosson et al., 2023). Although the underlying physics of glaciers is similar to that of ice sheets, glaciers are usually studied with different models because of their smaller spatial scales. Modern ice sheet models typically have resolutions of ∼1 to 10 km, whereas mountain glaciers require sub-km resolutions to accurately represent surface
and bed topography, mass balance, and ice-flow dynamics.

Glacier mass balance (MB), geometry evolution, and ice-flow dynamics modeling has seen significant advancements over the past decade (Zekollari et al., 2022b), with the availability of global glacier inventories (RGI Consortium, 2017), distributed global- and regional-scale ice thickness estimates (Farinotti et al., 2019; Grab et al., 2021; Millan et al., 2022), and satellite-derived mass-balance measurements (Hugonnet et al., 2021). The representation of glacier geometry has evolved from the
simplified volume-area/length scaling of early models (Bahr et al., 1997; Hock et al., 2019) to geometric models that dynamically adjust area and surface elevation at all elevations (Huss and Hock, 2015), and recently to two-dimensional flowline dynamics based on the shallow-ice approximation (e.g., the Open Global Glacier Model (OGGM v1.1; Maussion et al. 2019) and the Global Glacier Evolution Model flow (GloGEMflow; Zekollari et al. 2019)). Some newer developments include 3D geometry of individual glaciers (e.g., GloGEMflow3D (Zekollari et al., 2022a)). Advances in computational power and numerical
techniques, such as GPU processing and machine learning (ML) approaches, have further enabled 3D models of glacier evolution over long timescales with improved resolution. For example, the Instructed Glacier Model (IGM) applies deep learning to emulate Stokes dynamics and predict the evolution of ice sheets, ice caps, and glaciers (Jouvet et al., 2021; Jouvet and Cordonnier, 2023). Other ML applications for glacier modeling have also been explored, including MB reconstruction (Guidicelli et al., 2023), MB uncertainty estimation (Diaconu and Gottschling, 2024), and ice flow dynamics (Bolibar et al., 2023). How-
ever, applications of full-Stokes models (which solve the 3D Stokes equations for ice flow) to large regions and global-scale assessments have been limited due to computational requirements (Zekollari et al. (2022b), and the citations therein).

To provide a structured framework for comparing multiple glacier models, assessing their performance, and improving confidence in model projections, the Glacier Model Intercomparison Project (GlacierMIP) was launched in 2015. The first two phases of GlacierMIP focused on glacier evolution through the 21$^{st}$ century. GlacierMIP1 (2015-2019) compared published
projections from six glacier evolution models (Hock et al., 2019). None of the six models included prognostic ice dynamics; instead, five models relied on volume–area/length scaling for geometry change (Slangen et al., 2011; Radić et al., 2013; Marzeion et al., 2012; Hirabayashi et al., 2013; Giesen and Oerlemans, 2013), and one model used an empirical glacier evolution scheme (Huss and Hock, 2015). Eleven models took part in GlacierMIP2 (2019-2020) (Marzeion et al., 2020), including the flowline models OGGM and GloGEMflow. The third phase, GlacierMIP3 (Zekollari et al., 2024), built on the
first two phases and investigated the long-term equilibration of glaciers under constant climate conditions. In GlacierMIP3, the models have become increasingly sophisticated in terms of geometry representation, mass balance, and ice dynamics.

This study presents a new framework for simulating glaciers in the Community Ice Sheet Model (CISM), the ice dynamics component of the Community Earth System Model (CESM; Danabasoglu et al. 2020). Originally developed to simulate the evolution of the Greenland and Antarctic ice sheets (Lipscomb et al., 2019, 2021), CISM is the first 3D, higher-order ice-flow
model to participate in GlacierMIP. In contrast to the 2D flowline models, which resolve the flow only in the $x - z$ plane (where $x$ is the direction of motion), CISM is a dynamic model that prognoses the ice velocity, temperature, and stresses in



three dimensions. As a higher-order model, it includes not only vertical shear stresses, but also longitudinal and lateral stresses, in the ice momentum balance (Hindmarsh, 2004; Pattyn et al., 2008).

The new CISM developments support high-resolution, regional-scale glacier simulations, similar to other regional glaciation
models (Seguinot et al., 2018; Clarke et al., 2015). This contrasts glacier-centric models, such as GloGEMflow3D and OGGM, that model each glacier independently. When run as a glacier model, CISM can compute and calibrate the surface mass balance (SMB) for all glaciers in the domain, optimize the agreement between modeled and observed glacier area and thickness, and track the advance and retreat of each glacier.

In this work, we apply the new glacier-enabled model to all the glaciers of the European Alps within the framework of
GlacierMIP3. After reviewing CISM's ice-sheet capabilities (Sect. 2), we describe the new developments that support glacier simulations (Sect. 3). We present the model application to the European Alps, showing results of spin-up and equilibration experiments (Sect. 4). We then evaluate the computational performance (Sect. 5), discuss model limitations and future work (Sect. 6), and offer conclusions (Sect. 7).

## 2   CISM as an ice sheet model

CISM is a parallel, open-source code, written in Fortran and Python, which can be run either as a standalone ice sheet model or as a coupled component of CESM. As a standalone model, CISM has participated in several community comparisons, including the Ice Sheet Model Intercomparison Project for CMIP6 (ISMIP6; Nowicki et al. 2020; Goelzer et al. 2020; Seroussi et al. 2020), the Linear Antarctic Response to basal melting Model Intercomparison Project (LARMIP2; Levermann et al. 2019), and the Antarctic Buttressing Model Intercomparison Project (ABUMIP; Sun et al. 2020). As a coupled CESM component,
CISM has been used to study the evolution of the Greenland ice sheet in future warm climates (Muntjewerf et al., 2020a, b) and during the Last Interglacial period (Sommers et al., 2021).

Since the underlying physics of glaciers is similar to that of ice sheets, many of CISM's numerical algorithms can be used for glaciers without modifications. This section summarizes methods common to ice sheets and glaciers, while Sect. 3 describes the new glacier-related developments.

### 2.1   Dynamical core

CISM includes a dynamical core, Glissade, which solves conservation equations for mass, momentum, and thermal energy to determine changes in ice thickness, velocity, and internal temperature. The model runs on a structured rectangular grid with scalars (e.g., ice thickness and temperature) located at the cell centers and velocity components at cell corners.

The most complex part of the dynamical core is a velocity solver that incorporates a hierarchy of Stokes-flow approximations,
including (1) the shallow-ice approximation (SIA; Hutter 1983), (2) the shallow-shelf approximation (SSA; MacAyeal 1989), (3) a depth-integrated higher-order approximation (DIVA) based on Goldberg (2011), and (4) a higher-order approximation based on Blatter (1995) and Pattyn (2003).





With the Blatter-Pattyn (BP) approximation, CISM solves a 3D set of elliptic equations for the horizontal velocity components $(u, v)$ at all vertical levels. DIVA simplifies the problem by solving a 2D set of elliptic equations for the vertically averaged velocity components $(\bar{u}, \bar{v})$ in each ice column, and then integrating locally through each column to obtain the full 3D velocity profile. The DIVA solver computes velocities similar to the BP velocities in most glaciated regions but is computationally much faster than BP. DIVA also scales well to the high resolutions needed to model mountain glaciers (Robinson et al., 2022). We therefore used DIVA for the simulations in this study. DIVA solves the following approximation of the Stokes equations in the $x$ direction (the $y$ equation is similar):

$$\frac{1}{H}\frac{\partial}{\partial x}\left[2\bar{\eta}H\left(2\frac{\partial \bar{u}}{\partial x}+\frac{\partial \bar{v}}{\partial y}\right)\right]+\frac{1}{H}\frac{\partial}{\partial y}\left[\bar{\eta}H\left(\frac{\partial \bar{u}}{\partial y}+\frac{\partial \bar{v}}{\partial x}\right)\right]+\frac{\partial}{\partial z}\left(\eta\frac{\partial u}{\partial z}\right)=\rho_i g\frac{\partial s}{\partial x}, \tag{1}$$

where $H$ is the ice thickness, $\bar{\eta}$ is the vertical mean viscosity, $\rho_i$ is the density of ice, $g$ is gravitational acceleration, and $s$ is the surface elevation. The three terms on the left side of the equation describe longitudinal stresses, lateral stresses, and vertical shear stresses, respectively. These internal ice stresses balance the gravitational driving stress on the right hand side.

The viscosity is a nonlinear function of temperature and strain rate:

$$\eta \equiv \frac{1}{2}A^{\frac{-1}{n}}\dot{\varepsilon}_e^{\frac{1-n}{n}}, \tag{2}$$

where $A$ is a temperature-dependent flow factor, $\dot{\varepsilon}_e$ is the effective strain rate (derived from the 3D strain rate tensor), and $n = 3$ is the exponent in Glen's flow law. When solving Eq. (1), CISM has an option to cap the magnitude $\|\partial s/\partial x\|$ of the surface slope to a value of $m_{\mathrm{max}}$ to maintain model stability in regions of steep topography. After using Eq. (1) to compute the vertical mean velocity, the solver integrates in the vertical direction to find the full 3D velocity (see Goldberg (2011) and Lipscomb et al. (2019) for details). The ice velocity can be partitioned into a sliding velocity $u_b$ and an internal deformational velocity. CISM computes the deformational velocity as a function of the viscosity, which depends on the ice temperature and deformation rate; warmer and faster-deforming ice is softer.

The model supports several basal friction schemes with different relationships between sliding velocity and shear stress. For the simulations in this study, the sliding velocity is computed using a Weertman-type power law (Weertman, 1957):

$$\tau_b = C_p u_b^{1/m}, \tag{3}$$

where $\tau_b$ is the basal shear stress (a boundary condition for Eq. (1)), $u_b$ is the basal sliding speed, $m$ is a power-law exponent, and $C_p$ is a spatially varying friction coefficient.

CISM transports mass and internal energy using incremental remapping (Lipscomb and Hunke, 2004) – a conservative, second-order-accurate upwind scheme that preserves the monotonicity of ice temperature and other tracers. An implicit vertical solver computes interior heat dissipation and conduction. At the lower surface, CISM computes the temperature (if below the pressure melting point) or melt rate based on the balance of frictional, geothermal, and conductive heat fluxes.

## 2.2 Ice sheet initialization

Ice sheets are initialized in CISM by spinning up the model for thousands of simulated years until the ice geometry reaches a steady state consistent with the applied forcing. External forcing consists of the surface mass balance (which determines





thickness changes at the upper ice surface), the surface air temperature (an upper boundary condition for ice thermodynamics), the geothermal heat flux (a lower boundary condition for thermodynamics of grounded ice), and ocean thermal forcing in sub-ice-shelf cavities for floating ice. The climate forcing used during the spin-up is derived from observations or models and is typically from a period in the 20$^{\text{th}}$ century when the ice sheet was in approximate balance with the climate. During the spin-up, the ice sheet is nudged toward an observation-based thickness target (e.g., Morlighem et al. 2014, 2019). For grounded ice, this

is done by adjusting a spatial field of friction coefficients in the basal sliding law (Pollard and DeConto, 2012; Lipscomb et al., 2021).

The initialization is followed by a historical run to advance the ice sheet state to the present day, and then a projection run that continues into the future. Forcing during the historical run comes from recent observations and reanalyses, while the projection run uses output from simulations of future climate by regional models or Earth system models (ESMs).

## 3 CISM development for glacier modeling

To participate in GlacierMIP3, we modified CISM to simulate glacier evolution at regional scales. The goals of GlacierMIP3 were to (1) estimate the equilibrium area and volume of all glaciers outside the two ice sheets, if temperatures were to stabilize at present-day levels, (2) make similar estimates for temperature changes under various climate change scenarios, and (3) determine the time needed to reach a new equilibrium. The GlacierMIP3 protocol specifies that each glacier should be initialized

to its observed state at the RGIv6 date, usually around the year 2000.

Initializing a glacier requires calibration with respect to atmospheric forcing by tuning model parameters so that the initial state is consistent with the RGIv6 extent, observed mass balance, and ice thickness estimates. After initialization to the RGIv6 date, each glacier is run forward to equilibrium for either 2000 years or 5000 years, depending on the region. Further details on the protocols and the prescribed atmospheric forcing are available on the GlacierMIP3 GitHub page (https://github.com/

GlacierMIP/GlacierMIP3).

Table 1 shows the values of the parameters introduced below and used in glacier simulations.

### 3.1 Glacier identification and tracking

Each glacier in the RGIv6 inventory has a unique identification number. The first step to modeling these glaciers in CISM is to remap the RGI outlines to the CISM grid (Sect. 4.1) and assign an RGI ID to each grid cell with a non-zero ice thickness. At

startup, CISM makes a list of all the unique RGI IDs in the domain. It puts these in numerical order and associates each RGI ID with a CISM glacier ID between 1 and $N_g$, where $N_g$ is the total number of glaciers in the domain. Numbering the glaciers in order, without gaps, facilitates calculations that require looping over all glaciers.

Cells that are initially ice-free receive a CISM ID of 0. When a glacier retreats, any newly ice-free cell is also given an ID of 0. If the glacier re-advances to a cell from which it previously retreated, the initial ID is restored. Some glaciers will advance

into cells that were initially ice-free; a cell is deemed to be newly glaciated when its ice thickness $H$ exceeds a prescribed value, $H_{\text{min}}$. When this happens, CISM looks upstream to the grid cells that are ice sources for the new glacier cell. In most cases,





**Table 1.** Default values of various physical constants and glacier-specific parameters in CISM. These parameters are user-defined and can be adjusted during the initial model setup.

| Name | Value and units | Symbol |
|---|---|---|
| Ice density | 917 kg m$^{-3}$ | $\rho_i$ |
| Gravitational acceleration | 9.81 m s$^{-2}$ | $g$ |
| Glen's exponent | 3 | $n$ |
| Max slope for surface gradients | 0.2 | $m_{\mathrm{max}}$ |
| Geothermal heat flux | 0.05 Wm$^{-2}$ | G |
| Uniform atmospheric lapse rate | 6.0 $^{\circ}$C km$^{-1}$ | $\lambda$ |
| Melt threshold temperature | $-1^{\circ}$C | $\mathrm{T}_{\mathrm{melt}}$ |
| Snow–rain threshold temperatures | $0 - 2^{\circ}$C | $T_s^{\mathrm{min}}, T_s^{\mathrm{max}}$ |
| Ablation factor (initial/default) | 1500 mm w.e. $^{\circ}$C$^{-1}$ yr$^{-1}$ | $\mu_{\mathrm{init}}$ |
| Ablation factor (range) | $300 - 4000$ mm w.e. $^{\circ}$C$^{-1}$ yr$^{-1}$ | $\mu_{\mathrm{min}}, \mu_{\mathrm{max}}$ |
| Precipitation factor (initial/default) | 1.0 | $\alpha_{\mathrm{init}}$ |
| Precipitation factor (range) | $0.3 - 3$ | $\alpha_{\mathrm{min}}, \alpha_{\mathrm{max}}$ |
| Temperature correction (initial/default) | 0 $^{\circ}$C | $\beta_{\mathrm{init}}$ |
| Temperature correction (range) | -5 $- 5$ $^{\circ}$C | $\beta_{\mathrm{min}}, \beta_{\mathrm{max}}$ |
| Basal friction coefficient (initial/default) | $3.0 \times 10^4$ Pa (m yr$^{-1}$)$^{-1/n}$ | $C_p^{\mathrm{init}}$ |
| Basal friction coefficient (range) | $3.0 \times 10^3 - 1.0 \times 10^5$ Pa (m yr$^{-1}$)$^{-1/n}$ | $C_p^{\mathrm{min}}, C_p^{\mathrm{max}}$ |
| Thickness scale for basal tuning | 200 m | $H_0$ |
| Time scale for basal tuning | 200 yr | $\tau_0$ |
| Relaxation factor for basal tuning | 0.05 | $f_r$ |
| Redistribution rate for advanced ice | 1 m yr$^{-1}$ | $R$ |
| Minimum ice thickness | 1 m | $H_{\mathrm{min}}$ |

the upstream ice belongs to a single glacier, which provides the ID for this cell. If the upstream ice belongs to two different glaciers, CISM must choose between them. We determined that selecting the upstream glacier ID that yields a more negative SMB for the new glacier cell is optimal, as a negative SMB inhibits further advance beyond the original RGI boundary.

## 155  3.2 Surface mass balance

When run as an ice sheet model, CISM usually does not compute the SMB internally. Instead, the SMB is an input calculated offline by a regional climate model or computed at runtime by the land component of CESM when interactively coupled. For CISM as a glacier model, coupling between the land and land-ice components of CESM has not been implemented. Therefore, we use a simple temperature-index method to calculate the SMB, similar to the scheme in OGGM (Maussion et al., 2019).



In this scheme, CISM computes the SMB of glaciers based on the monthly mean surface air temperature and precipitation rate, remapped to the CISM grid (see Sect. 4.2). The input air temperature usually is provided at the reference elevation of the coarse-resolution atmospheric forcing. CISM downscales the air temperature from the reference elevation to the local surface elevation on the (high-resolution) CISM grid based on a fixed lapse rate $\lambda$ (Table 1). The fraction of total precipitation reaching the glacier surface as snow depends on the downscaled temperature $T$. Precipitation is assumed to fall entirely as snow when $T <= T_s^{\min}$ and as rain when $T >= T_s^{\max}$. At temperatures between $T_s^{\min}$ and $T_s^{\max}$, the snow fraction varies linearly between 1 and 0.

The SMB is calculated from the difference between the snowfall and the melt, which is calculated from a temperature-index scheme:

$$B_i = \alpha S_i - \mu \max(T_i - T_{\text{melt}}, 0), \tag{4}$$

where $B_i$ is the SMB for grid cell $i$, $S_i$ is the snowfall rate, $T_i$ is the surface temperature, and $T_{\text{melt}}$ is a temperature threshold for melting. The quantities $B_i$, $S_i$, and $T_i$ are monthly means at a given location. The units of $B_i$ and $S_i$ are mm water equivalent (w.e.) $\text{yr}^{-1}$, and $T_i$ is measured in deg C. Within the model, the ablation factor $\mu$ (also known as degree-day factor, melt factor, or temperature sensitivity parameter) is computed in mm w.e. $°\text{C}^{-1}$ $\text{yr}^{-1}$ while the precipitation correction parameter $\alpha$ is dimensionless.

The calibration parameters $\mu$ and $\alpha$ are determined for each individual glacier in a two-step calibration process:

1. We assume that the glacier was in a state of approximate balance with the climate during a baseline period in the 20[th] century (i.e., the glacier-specific mean SMB was zero):

$$B = \sum_{m=1}^{12} \sum_{i=1}^{N} [\alpha S_{im} - \mu \max(T_{im} - T_{\text{melt}}, 0)] = 0, \tag{5}$$

where $B$ is the annual mean SMB for the glacier, the first summation is over the months of the year, the second summation is over the $N$ grid cells in the glacier, and the subscript $m$ denotes a particular month. For a given value of $\alpha$ (e.g., $\alpha = 1$ if the precipitation is unbiased), we can sum over all months and grid cells in Eq. (5) and obtain $\mu$ for each glacier.

2. If we have observations of both SMB and atmospheric forcing during a recent period when the glacier was out of balance with the climate, we introduce a second criterion. For each glacier we supplement Eq. (5) with a similar equation for the recent period:

$$\hat{B} = \sum_{m=1}^{12} \sum_{i=1}^{N} \left[ \alpha \hat{S}_{im} - \mu \max(\hat{T}_{im} - T_{\text{melt}}, 0) \right], \tag{6}$$

where the carets denote quantities taken over the recent period.

Equations (5) and (6) form a system of two equations with two unknowns, which can be solved for $\alpha$ and $\mu$ for each glacier. The summation over $N$ cells for each glacier includes all the cells belonging to the glacier based on the RGI outlines, including ice-free cells bordering the glacier (within one cell), provided they are in the ablation zone. In calculating the glacier's





area-integrated annual-mean SMB, we include these neighboring cells because ice flows into and melts within them through-
out the year. If these cells were excluded, the temperature sums in Eqs. (5) and (6) would be too small, and $\mu$ would be
overestimated as a result.

The parameters $\alpha$ and $\mu$ are required to fall within physically reasonable ranges $(\alpha_{\min}, \alpha_{\max})$ and $(\mu_{\min}, \mu_{\max})$. For some
(usually small) glaciers, the $\alpha$ and $\mu$ computed from Eqs. (5) and (6) can lie outside these ranges because of atmospheric
forcing biases or observational errors in $\hat{B}$. In such cases, the first option is to ignore Eq. (6), set $\alpha$ to its default value of 1.0,
and solve Eq. (5) for $\mu$. This is done, for example, if $\hat{B} > 0$ (i.e., the glacier is supposedly gaining mass) in a warming climate.
If this first option fails to yield $\mu$ within the defined range, a second option is to introduce a temperature correction $\beta$, which
is added to $T_i$ for each cell in the glacier. This option is needed if $T$ has a strong cold bias, resulting in little or no ablation
even when $\mu$ is large. In Sect. 4.3, we discuss the distribution of calibrated values within the $\alpha$, $\mu$, and $\beta$ ranges (Table 1) for
glaciers in the Alps.

### 3.3 Glacier initialization and thickness inversion

To initialize glaciers, we use a procedure similar to the method described for ice sheets in Sect. 2.2. That approach, however,
relies on the area and thickness targets being appropriate for a steady state in equilibrium with the climate. Most glaciers have
been losing mass in recent decades and are out of balance with the climate (Zemp et al., 2019; Zekollari et al., 2020), and
therefore estimates of present-day or recent ice thickness (e.g., Farinotti et al. 2019) are not a suitable target for a steady-state
spin-up.

We therefore divide the initialization into two steps. The first step is a spin-up run of several thousand years, with climate
forcing corresponding to a baseline year (e.g., in the mid to late 20[th] century) when glaciers are assumed to be approximately
in equilibrium with the climate. The second step is a historical run from the baseline year to the outline date for glaciers in
the RGIv6 dataset. During the historical period, most simulated glaciers lose mass, consistent with the observational record.
The goal of the spin-up is to initialize each glacier with an extent and thickness corresponding to the baseline year. If we lack
region-wide observations for this year (as is usually the case), we make two approximations:

1. The retreat between the baseline year and the RGI year (2003) is relatively small; thus, we can use the unmodified RGI
   outlines as an area target. This assumption is undesirable for glaciers that retreated substantially between the baseline
and RGI dates, but we considered it preferable to guessing the earlier outlines without observational support.

2. The decrease in thickness at a specific location from the baseline year to the RGI year can be estimated by calculating
   the SMB between these two years. We accordingly make adjustments to the thickness, assuming that the SMB changes
   linearly over time and that the time between the baseline and RGI year is relatively short compared to the glacier's
   dynamic response time.

With the second assumption, we can adjust the baseline thickness targets at runtime to be consistent with the calibrated
historical SMB (Sect. 3.2). In a warming climate, the total ice volume will be greater at the baseline date than at the RGI date.
When the model is run forward with the historical SMB, the goal is to reach the observed volume at the RGI date, as specified





in the GlacierMIP3 protocol. For the runs described in Sect. 4, the baseline year is 1984, implying a 19-year historical run to 2003.

At the basal boundary, we assume an upward geothermal heat flux $G$. We compute the basal shear stress as a function of the sliding speed using Eq. (3). Since the friction coefficient $C_p$ is not well constrained, we use it as a tuning parameter to nudge the ice thickness in each grid cell toward the baseline target. The value of $C_p$ is initially set to $C_p^{\text{init}}$ and is constrained to lie in the range $(C_p^{\text{min}}, C_p^{\text{max}})$ (Table 1). Where the ice is thicker than its target value, $C_p$ is reduced to increase sliding and promote thinning, whereas $C_p$ is increased if the ice is thinner than its target.

Following Lipscomb et al. (2021), the rate of change of $C_p$ is given by

$$\frac{dC_p}{dt} = C_p \left[ \frac{(H_{\text{obs}} - H)}{H_0\,\tau_0} - \frac{2}{H_0}\frac{dH}{dt} - \frac{f_r}{\tau_0}\ln\left(\frac{C_p}{C_p^{\text{init}}}\right) \right], \tag{7}$$

where $H_{\text{obs}}$ is the observation-based thickness target, $H_0$ is a thickness scale, $\tau_0$ is a relaxation time scale, and $f_r$ is a relaxation factor. The first term on the right side of the equation minimizes differences between $H$ and $H_{\text{obs}}$; the second term dampens oscillations in $C_p$ as $H$ approaches $H_{\text{obs}}$; and the last term prevents $C_p$ from drifting toward the max or min value when the

first term is small but non-zero.

In most regions, this tuning procedure yields a steady-state ice thickness close to the target value. The procedure cannot, however, add ice to grid cells that are ice-free for other reasons (e.g., a highly negative SMB). Nor can it remove ice from cells that advance beyond the RGI outlines, but it gives these cells a low value of $C_p$ that reduces the thickness. An alternate approach, often used in SIA glacier models, is to tune the softness parameter $A$ in the ice viscosity, Eq. (2) (Zekollari et al.,

2022b). In our CISM simulations, $A$ is not a tuning parameter but is determined internally by the evolving ice temperature.

We found that during initialization, glaciers tend to advance to a steady state beyond the RGI outlines. The main reason for this advance is that the transport scheme allows some ice to diffuse into previously ice-free grid cells in the accumulation zone, where it is not removed by melting. To limit this spurious advance, we impose two corrections. In advanced cells (i.e., cells that are ice-covered in the model but are ice-free in the RGI data), we allow ablation where the computed SMB is negative, but we

forbid accumulation where the computed SMB is positive. At the same time, we remove ice from advanced cells at a rate $R$ and spread this ice, conserving mass, over the target area of the glacier. This can be viewed as a crude model of mass redistribution by avalanches at glacier walls. A redistribution rate $R = 1$ m yr$^{-1}$ is enough to significantly reduce glacier advance. We do not, however, apply a negative SMB to advanced grid cells in the accumulation zone, since this would be a nonphysical sink for ice mass.

## 4   CISM application to the European Alps

### 4.1   Model domain and resolution

The CISM domain for this study covers the European Alps (Figure 1), with 3892 individual glaciers (out of 3927 glaciers in RGIv6 region 11). These glaciers cover a combined area of ∼2089 km$^2$. We excluded 35 small glaciers in the Pyrenees, Montenegro, and Albania (combined area of ∼3 km$^2$, i.e. ∼0.1% of the regional area) from the original RGIv6 region 11 to





**Figure 1.** Top: CISM domain for the European Alps, which contain most of the glaciers in RGI region 11 (Central Europe). Shading shows the surface elevation profile (m). Bottom: Input ice thickness (m) from Farinotti et al. (2019), remapped to the 100-m CISM grid, for four sub-domains (A–D).

reduce the model domain size. The two largest glaciers in the region are the Aletsch ($\sim$82 km$^2$) and Gorner ($\sim$56 km$^2$), while more than 40% of the glaciers have areas smaller than 0.05 km$^2$. Therefore, the model grid must be fine enough to accurately represent these small glaciers and their ice dynamics.

We created two grids with horizontal resolutions of 100 m and 200 m, using the coarser grid for model development and the finer grid for GlacierMIP3 production runs. At 100-m resolution, six glaciers are at sub-grid scale, resulting in 3886 glaciers represented on the grid (Table 2). Of these, 42 glaciers occupy one grid cell each.





## 4.2 Forcing and initialization data

CISM requires five types of input data for glacier simulations: the outline and ID of each glacier, continuous surface elevation over the domain, ice thickness, atmospheric forcing (monthly mean temperature and precipitation), and glacier geodetic mass balance. All data were mapped to the 100-m and 200-m CISM grids.

1. We used the glacier outlines and IDs from RGIv6. For the Alps, the vast majority of glacier outlines are from summer 2003. We refer to this year as the RGI date and take it as the reference year for the glacier outlines.

2. To create continuous surface elevation, we used two Digital Elevation Model (DEM) sources: (1) the USGS 3 arc-sec (90-meter) SRTM (Shuttle Radar Topography Mission) DEM and (2) finer-resolution glacier-specific surface DEM tiles taken from Farinotti et al. (2019) (hereafter F19). We merged these two sources, with the latter taking precedence in
overlapping regions.

3. The distributed ice thickness was also taken from the F19 five-model consensus estimate.

4. The atmospheric forcing was taken from W5E5 v2.0 data from the Inter-Sectoral Impact Model Intercomparison Project phase 3b (ISIMIP3b) at daily, 0.5° resolution from January 1979 through December 2019 (Lange et al., 2021). This is a merged dataset with WATCH Forcing Data methodology applied to ERA5 data (Cucchi et al., 2020). The variables
needed for CISM are near-surface air temperature (°C), total precipitation flux (kg m$^{-2}$ s$^{-1}$), and surface altitude of the data (m). We averaged the atmospheric forcing to monthly mean values.

5. We took the glacier-wide geodetic mass balance from Hugonnet et al. (2021).

Since the W5E5 v2.0 forcing data were available only from 1979 onward, we used 1984 as the baseline year and constructed a baseline climatology by computing the 1979–1988 mean, assuming that glaciers were in approximate balance during this
decade. We constructed a recent climatology (nominally for year 2010) by taking the 2000–2019 mean, the same period over which Hugonnet et al. (2021) calculated the geodetic mass balance used for SMB calibration (Sect. 3.2).

To calibrate the SMB at runtime, we apply the baseline climatology in Eq. (5) and the recent climatology in Eq. (6). During the historical run from the baseline date to the RGI date, we interpolate between the baseline and recent climatology, assuming that monthly mean temperature and precipitation evolved linearly in time between 1984 and 2010.

## 4.3 Spin-up and mass balance calibration

We conducted a 10,000-year spin-up for the Alps. For the first 8,000 years, we invert for the basal friction parameter $C_p$ in Eq. (7) as described in Sect. 3.3. For the last 2,000 years, we hold $C_p$ fixed to avoid initialization shocks in the forward runs. Throughout the spin-up, we nudge each glacier towards its RGI-based area target using the calibration procedure described in Sect. 3.2.

The SMB parameters $\mu$ and $\alpha$ in Eq. (4) are calibrated for each glacier to yield a balanced state for the 1979-1988 climate while matching the Hugonnet et al. (2021) geodetic mass balance over 2000–2019. Figure 2 shows the distribution of these



**Table 2.** Cumulative statistics for glaciers in the European Alps for the input data, at the end of the spin-up (1984), and at the RGI date (2003).

| | |
|---|---|
| Number of input glaciers on 100-m CISM grid | 3886 |
| Input glacier area (RGIv6) | 2086.4 km$^2$ |
| Input glacier volume (F19) | 126.6 km$^3$ |
| Target glacier area (baseline year of 1984) | 2086.4 km$^2$ |
| Target glacier volume (baseline year of 1984) | 138.9 km$^3$ |
| Post spin-up glacier area (~1984) | 2500.9 km$^2$ |
| Post spin-up glacier volume (~1984) | 138.0 km$^3$ |
| Glacier area after historical run (~2003) | 2250.0 km$^2$ |
| Glacier volume after historical run (~2003) | 127.0 km$^3$ |

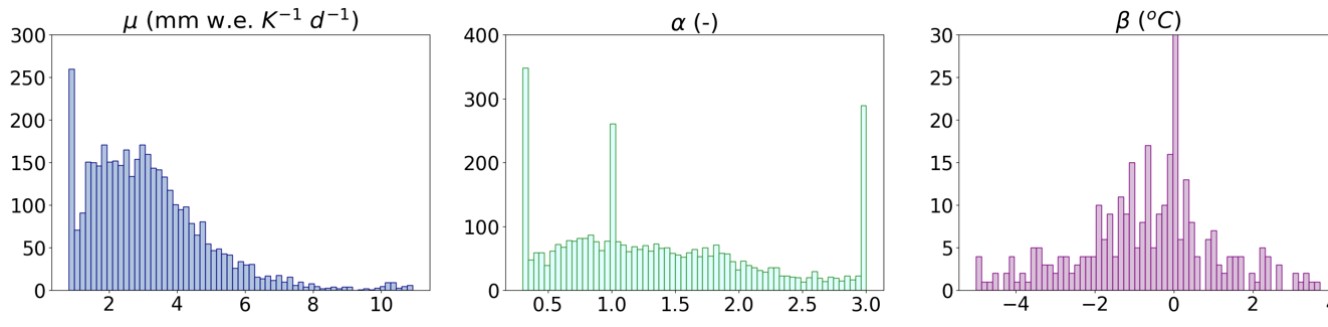

**Figure 2.** Distribution of the SMB parameters described in Sect. 3.2 for all simulated glaciers in the Alps. The x-axis shows the range of values for each parameter (Table 1), and the y-axis shows the number of glaciers. The y-axis in panel c is capped at 30, because most glaciers (n=3593) have $\beta$=0.

parameters and the $\beta$ correction (Sect. 3.2) for all glaciers in the domain. The $\mu$ values are limited to a range between 300 and 4000 mm w.e. °C$^{-1}$ yr$^{-1}$ (approximately $0.8 - 11$ mm w.e. °C$^{-1}$ day$^{-1}$), and $\alpha$ to a range between 0.3 and 3 (Table 1). The correction $\beta$ can be of either sign and is limited to a magnitude of 5 °C. These ranges are similar to, but somewhat narrower than, those in OGGM assessments. For example, Schuster et al. (2023) set the corresponding $\mu$ ranges to $0.33 - 33$ mm w.e. °C$^{-1}$ day$^{-1}$, $\alpha$ to $0.1 - 10$, and $\beta$ to $-8 - 8$ °C.

For glaciers in the Alps, $\mu$ has a median value of 2.9 and a mean of 3.2 ±1.7 mm w.e. °C$^{-1}$ day$^{-1}$ across all glaciers (without weighting according to glacier area or volume). More than 95% of the glaciers have $\mu < 6.3$, which aligns with previous literature. For example, Hock (2003) reported this parameter (referred to as the degree-day factor) ranging from 5–12 mm w.e. °C$^{-1}$ day$^{-1}$ for individual glaciers globally, while Braithwaite and Hughes (2022) reported values ranging from 4.1–6.8 for eight glaciers in the Alps. Schuster et al. (2023) assessed various temperature-index models and calibration methods and found a range for calibrated $\mu$ of ~4–10 mm w.e. °C$^{-1}$ day$^{-1}$ for 88 glaciers globally.




The median value of $\alpha$ is 1.22 (Figure 2b). There are 303 glaciers with the lower threshold value of 0.3 and 270 glaciers with the upper value of 3.0, suggesting that some of these glaciers require a temperature correction $\beta$. We find that 293 glaciers

have nonzero $\beta$ (Figure 2c), but all have values well within the $\pm\,5\,°C$ threshold. Several factors, including variations among different glacier tributaries, as well as biases in climate forcing and the geodetic mass balance data, influence the ranges of these glacier-specific parameters.

## 4.4 Ice thickness and extent

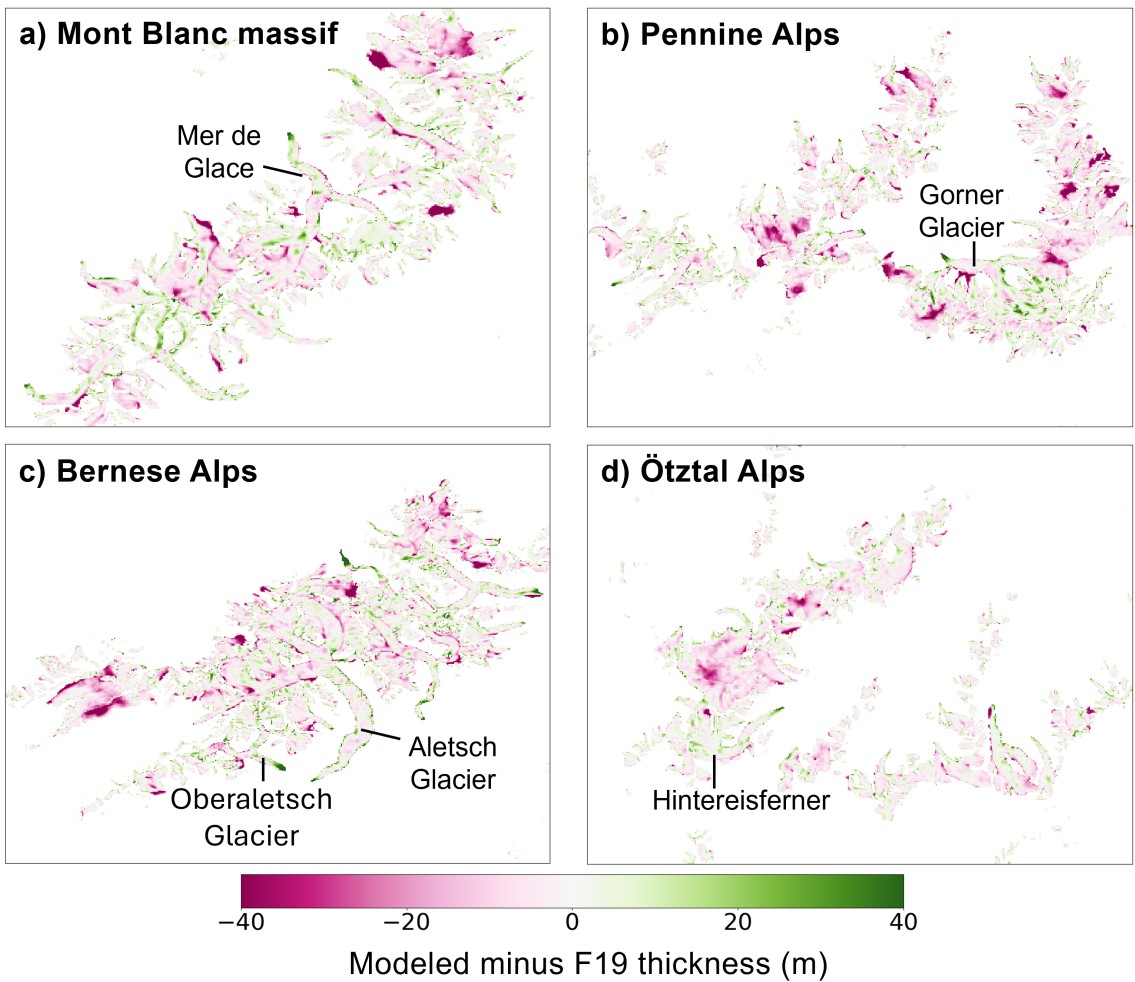

**Figure 3.** Difference between CISM and F19 ice thickness (m) for the four sub-domains in Figure 1. Green colors correspond to greater thickness in the CISM simulations.



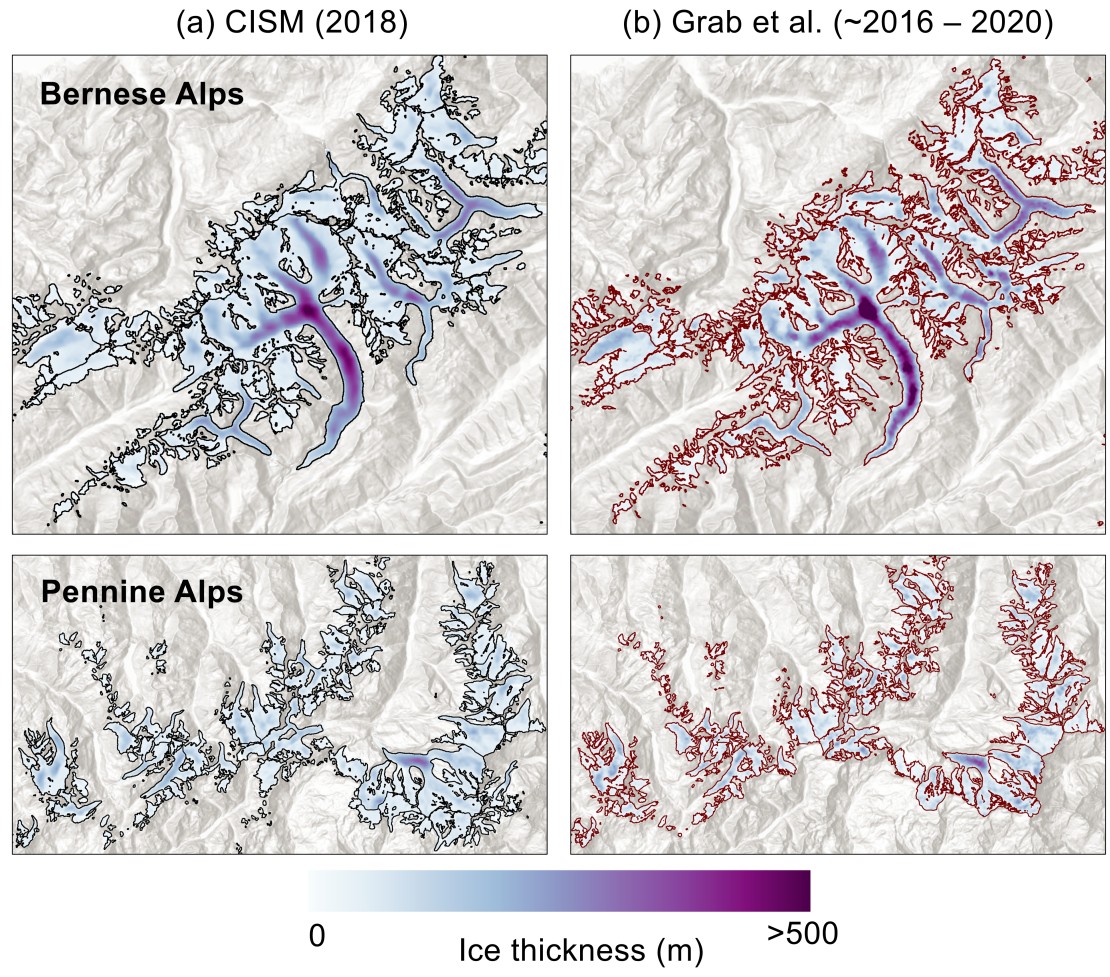

**Figure 4.** Comparison of CISM-simulated ice thickness (m) to observed thickness from Grab et al. (2021) for the Swiss Bernese and Pennine Alps around 2018. Note that the model and observations have different glacier extents in black and brown outlines, respectively.

We invert for ice thickness during the spin-up with the goal to match a baseline (1984) thickness target that is derived from
the F19 estimates. This target in many regions is thicker than F19 because of the SMB correction described in Sect. 3.3. The
resulting ice volume at the end of the spin-up (1984) is 138.9 km$^3$, about 10% larger than the F19 volume of 126.6 km$^3$
(Table 2). During the historical run to the RGI date, the volume decreases to 127.0 km$^3$, closely matching the F19 value. The
RMS error in ice thickness between the modeled estimate and F19 target across the entire domain is 14.8 m. In most cases,
positive differences (where the CISM thickness exceeds F19) occur along glacier peripheries and at termini (Figure 3). There
are some regions where the modeled ice is thinner than F19, which we discuss further in Sect. 6.

We also ran the model forward from the RGI date to 2018, to compare the modeled 2018 ice thickness with the Grab et al.
(2021) thickness measurements in the Swiss Alps, which used aerial ground-penetrating radar to determine ice thickness during



2016–2020. A direct comparison between the two datasets is not possible, since CISM extent is simulated while the Grab et al. (2021) extent is observed (hence the two have different glacier outlines), but we found similar magnitudes and spatial patterns in thickness (Figure 4).

The glaciers simulated in CISM have a total area of 2500.9 km$^2$ in 1984, much greater than the target area of 2086.4 km$^2$. When the model runs forward to 2003, the total area drops to 2250 km$^2$, closer to, but still above, the RGI targets (Table 2). We attribute the area excess to two main factors. First, there is some numerical diffusion of ice thickness under transport. This can lead to one or two rows of advanced ice, usually no more than a few meters thick, at glacier peripheries (Figure 3). Second, the RGI outlines have extensive ice-free regions at high elevations because of steep terrain that does not allow snow to accumulate. In CISM, ice can flow into these areas and remain; the "avalanche" redistribution described in Sect. 3.3 removes some but not all peripheral ice.

For individual glaciers, the mean difference in glacier area between RGIv6 and the 2003 model estimate is 0.04 ± 0.15 km$^2$ (mean ± std). For 17 large glaciers, the simulated area exceeds the observed area by more than 1 km$^2$. Glaciers with large differences include the Aletsch Glacier (area difference of 4.28 km$^2$), the Gorner Glacier (2.80 km$^2$), and the Oberaletsch Glacier system (2.56 km$^2$). As these glaciers have large accumulation zones and/or long tongues, the area increase is due to glacier advance into peripheral grid cells of the initial extent (Figure 3).

## 4.5 Surface velocity

We compared the modeled surface velocities with two satellite-derived estimates: ITS-LIVE (Gardner et al., 2022) and Millan et al. (2022), both remapped to the CISM grid (Figure 5). Although velocity is not an inversion target, the simulated surface velocities are in good agreement with observations, showing that CISM is capturing key processes governing ice flow.

Figure 5 shows simulated and observed velocity patterns for several large glaciers, where velocity differences can potentially be more prominent due to the greater surface area and thickness. Because the Millan et al. (2022) data spans 2017/2018, we limited our comparison of the three datasets to 2017, although seasonal glacier velocity changes may cause temporal misalignment in the three datasets. The model captures both the spatial patterns and overall magnitudes quite well. However, the model's accuracy varies for larger glaciers, particularly in accumulation zones and glacier tongues where it tends to overestimate velocities, unlike higher elevations and upper tributaries where it underestimates them (Figure 5b,c). Statistically, the differences between CISM and the two datasets are small. Figure 6a and b illustrate this for ITS-LIVE and Millan et al. (2022), respectively, focusing on the largest glaciers within the area that also have the greatest velocity differences. For cell-by-cell comparison, the mean differences are $20 \pm 33$ and $11 \pm 32$ m yr$^{-1}$ for the two datasets. Only a small fraction of the glacier area ($\sim$10%) has a difference exceeding $\pm 50$ m yr$^{-1}$. There could be several reasons for these differences. In general, velocity errors are correlated with thickness errors. Where the model underestimates (overestimates) ice thickness, the driving stress and the ice speed are likely to be to too low (high).



**Figure 5.** (a) CISM-simulated surface velocity (m yr$^{-1}$) for select large glaciers in the Alps. (b and c) Difference between CISM and two satellite-derived estimates; ITS-LIVE and Millan et al. (2022). All three velocity profiles are around the year 2017.

## 4.6 Committed ice loss

We assessed the committed ice loss for these glaciers, i.e., the long-term equilibrium ice loss if recent climate conditions (2000–2019) were to remain unchanged. For the commitment run, we started with the baseline state of 1984 and gradually introduced warming from 1984–2010 using a linear ramp. After 2010, we held the atmospheric forcing constant and continued the simulation for several centuries until the mass stabilized.

Figure 7 shows snapshots for the Swiss Bernese Alps (including the Aletsch Glacier) for four dates: the 1984 baseline, 2021,
2084 (100-years after the baseline), and 2184. By 2184, the glaciers are close to equilibration. Even in this highly optimistic climate scenario, there is significant area and volume loss (Figure 7d), emphasizing that glaciers are far from equilibrium with



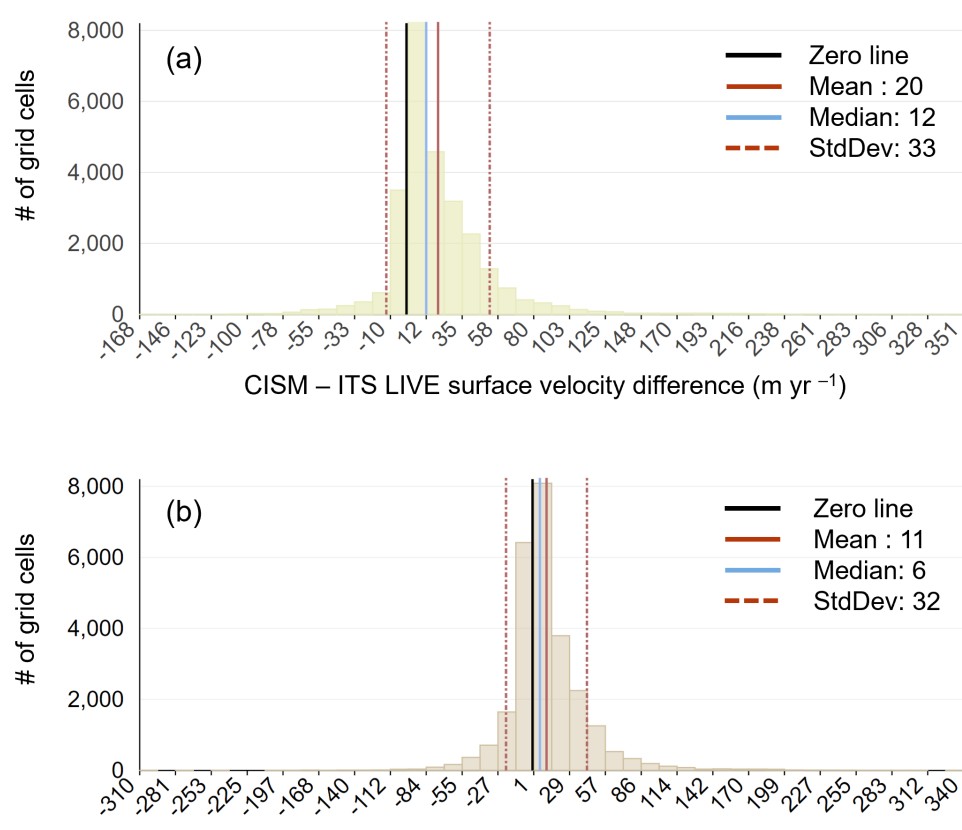

**Figure 6.** Velocity difference statistics between CISM and (a) ITS-LIVE and (b) Millan et al. (2022), around the year 2017. These are cell-by-cell differences for the 16 largest glaciers in the domain with volume > 1100 km$^3$ and area > 14 km$^2$. The limits of the x-axis correspond to the maximum difference.

the current climate. During this 200-year commitment run, the total ice volume in the entire Alps domain drops from 138 km$^3$ (1984) to 127 (2003), 104 (2021), 63 (2084), and 51 (2184). The total area drops from 2500 km$^2$ in 1984 to 994 km$^2$ by 2184.

These results are similar to those of Jouvet and Huss (2019), who studied the retreat of the Aletsch Glacier using a full-Stokes ice dynamics model. They found that, by 2100, the Aletsch Glacier is already committed to losing nearly half its 2017 volume based on the 2008-2018 climate, with a 32% loss projected using the 1988–2018 climate. CISM projects comparable volume loss with the 2000–2019 climate; approximately 35% between 2017 and 2100. We note that by 2017, the simulated Aletsch glacier has already lost 11% of its 1984 baseline volume of 14.27 km$^3$.




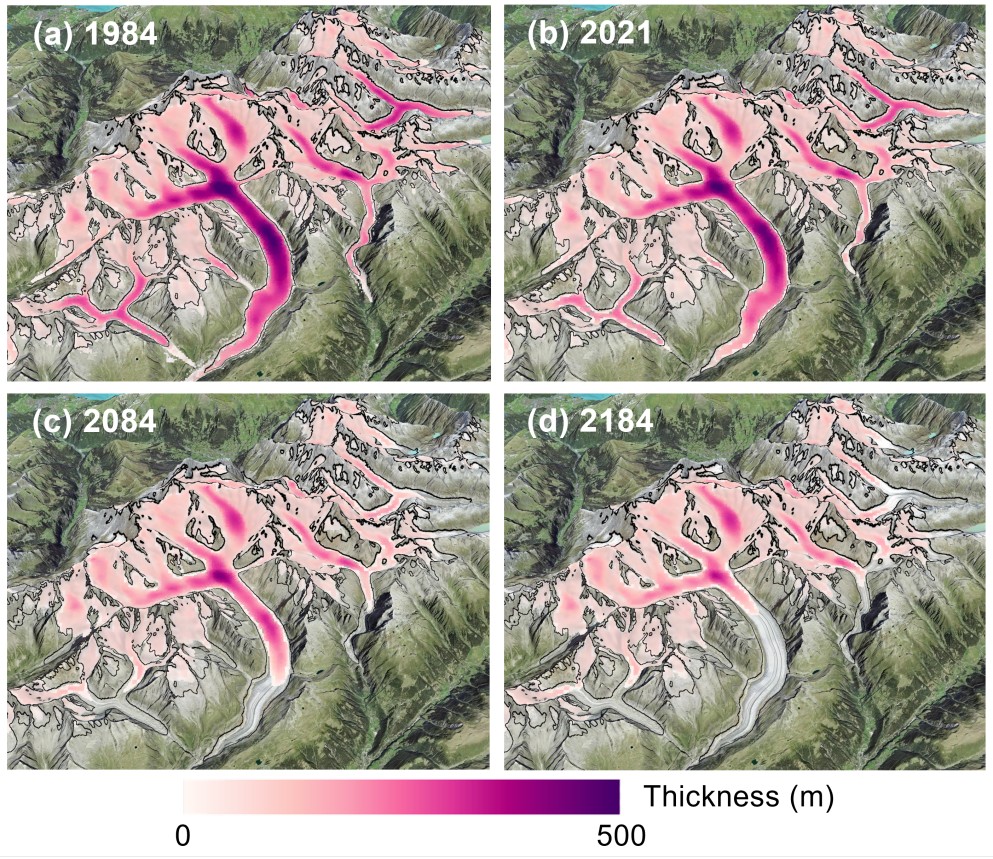

Thickness (m)

0                              500

**Figure 7.** Committed ice loss for the Bernese Alps. Ice thickness (m) for (a) the baseline climate of 1984, and (b–d) years 2021, 2084, and 2184, assuming continuation of recent climate (2000–2019) with no further warming. RGIv6 glacier outlines (∼2003) are shown in black.

## 4.7 Equilibration runs for GlacierMIP3

We carried out the full suite of GlacierMIP3 experiments for the Alps glaciers in RGI region 11 (Figure 1). Climate forcing for these experiments comes from five bias-corrected CMIP6 models from ISIMIP3b (GFDL-ESM4, IPSL-CM6A-LR, MPI-ESM1.2-HR, MRI-ESM2.0, and UKESM1-0-LL), for both the historical period (1850-2014) and the future. The future forcing includes three scenarios with different levels of warming: SSP126, SSP370, and SSP585. The forcing for an equilibrium run corresponds to one of eight 20-year periods as simulated by each of these CMIP6 models. The 20 years of data in each time

series are shuffled so that the forcing can be applied repeatedly for thousands of years without introducing spurious 20-year cycles (see the GlacierMIP3 GitHub page for more details).

     Our simulations show that Alpine glaciers will lose a large fraction of their area and volume under current temperatures, with near-total ice loss under warmer scenarios. For the 1995–2014 historical climate (Figure 8a), the cumulative volume loss for these glaciers is 56–63%, depending on the forcing dataset, and it will take 140–150 years to reach the equilibrium state



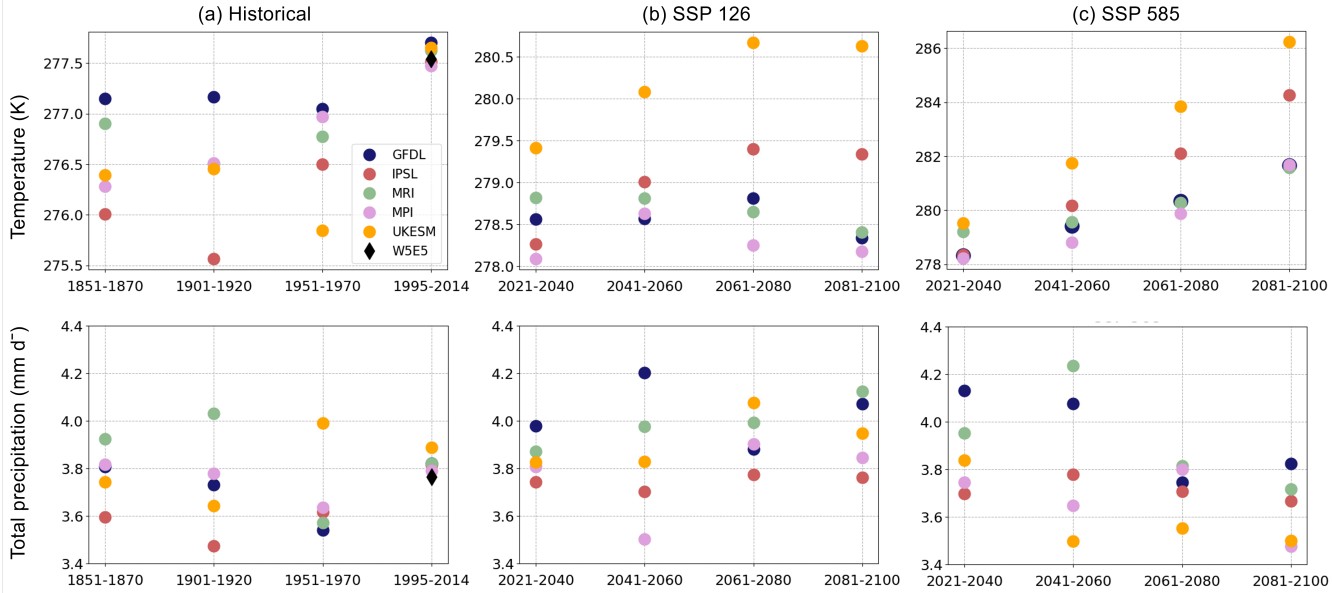

**Figure 8.** Temperature (K) and precipitation (mm d$^{-1}$) magnitudes in five CMIP6 models averaged over the glaciated grid cells in the Alps for the (a) historical, (b) SSP126, and (c) SSP585 scenarios. These are 20-year averages for the periods shown on the x-axis.

(i.e., when the slope of the trend-line is nearly flat, rounded to the nearest 5 years; Figure 9d). Under the sustainability-focused SSP126 scenario, volume loss ranges between 74% for MPI and 94% for UKESM with 2021–2040 climate forcing (Figure 9e). For this forcing, UKESM is 1.3°C warmer than MPI but has similar precipitation magnitude (Figure 8b).

For 2021–2040 forcing under SSP585, the domain-mean annual temperature ranges from 278.2 K (MPI) to 279.5 K (UKESM) (Figure 8c), and the total volume loss ranges from 73% (GFDL) to 94% (MRI and UKESM) (Figure 9i). In com-
parison to the other models, the GFDL forcing drives less volume loss due to the higher precipitation and lower temperature (Figure 8c). With slower warming, there is an increased response time; the time to equilibrium is about 140 years for GFDL, compared to 130, 100, and 90 years for MPI, IPSL, and UKESM/MRI, respectively; Figure 9i).

In the projected climate of the later decades of SSP585, CISM predicts a near-complete loss of area and volume for all five models, with an equilibrium response time of less than 50 years (Figure 9k and l). For the SSP126 scenario in the later decades,
MPI and GFDL show relatively less volume loss (about 75%; Figure 9h), noting that these models simulate slightly cooler temperatures than the other models (Figure 8b).

We refer the reader to Zekollari et al. (2024) for a detailed analysis of GlacierMIP3 simulations by all participating models.

## 5 Computational performance

3D glacier models are computationally expensive compared to their 2D counterparts, limiting the ability to model large
glaciated regions at high spatial resolutions on century-to-millennial scales. A twofold reduction in grid spacing quadruples the



**Figure 9.** CISM GlacierMIP3 experiment results, showing the percentage change in total ice volume in the Alps, for the climate periods and scenarios in Figure 8. The x-axis shows the first 300 simulation years of the 2000-year model run.

number of grid cells and typically halves the maximum stable time step, increasing the overall computational cost by a factor of eight. This suggests that the cost of glacier simulations using sub-kilometer grids would be several orders of magnitude higher than that of ice-sheet simulations, which with CISM are generally run at 4-km resolution. However, we have implemented computationally efficient schemes in CISM to make glacier runs more affordable.





## 5.1 Spatio-temporal resolutions and computational efficiency

For the Alps domain (Figure 1), we created grids at 100-m and 200-m resolutions to evaluate the sensitivity of model results and computational costs. The 100-m grid has dimensions of $9611 \times 5545$ (more than 50 million grid cells), of which only about 250,000 are glaciated.

To conserve computing resources, we created a glacier mask to identify regions that are currently ice-covered, or could potentially contain ice if glaciers advanced. The mask starts with ice-filled grid cells from the RGI input grid and is extended to a radius of 1 km around each glaciated cell. Upon initialization, CISM identifies and retains data blocks containing at least one cell within the glacier mask, labeling these as active while discarding the remaining blocks. We modified CISM's parallel routines (halo updates, gather/scatters, global sums, and broadcasts) to operate on active blocks only. This masking reduces the number of active blocks on the 100-meter Alps grid to fewer than 900, a tenfold reduction compared to running the simulation with inactive blocks included.

The atmospheric forcing consists of monthly mean temperature and precipitation. We found that runs on both grids were stable with a time step of 1 month (with each month containing exactly 30 days). This is similar to the maximum stable time step for CISM ice-sheet simulations on a 4-km grid. Ice sheets require shorter time steps relative to the grid spacing because the fastest outlet glaciers have speeds of several km yr$^{-1}$. In the Alps, the maximum glacier speeds are only several hundred m yr$^{-1}$ (Figure 5; Millan et al. 2022).

## 5.2 High-performance computing costs

On the 100-m grid with a time step of 1 month, we achieved throughput of about 140 model years per wallclock-hour on 896 processor-cores (7 nodes with 128 cores each) on Derecho, a high-performance computing system at the NSF National Center for Atmospheric Research. The cost per simulated year is about 6.4 cpu-hours. On the 200-m grid, with four times fewer grid cells, the throughput is about 430 model years per wallclock hour on 768 cores, a cost of about 1.8 cpu-hours per simulated year. Thus, the cost per cpu-hour on the 100-m grid is less than 4 times the cost on the 200-m grid. This superlinear scaling is a result of the method used to discard inactive blocks. Because each block in the 100-meter grid covers a smaller area than the 200-meter grid, there are far fewer active blocks than inactive ones.

These numbers suggest that applying CISM to regions larger than the Alps is computationally practical, particularly with a coarser resolution of 200 m. The four largest RGI regions (Alaska, Arctic Canada North, the Greenland periphery, and Antarctic islands) have a glaciated area of $\sim 10^5$ km$^2$ each. At 200-m resolution, this would imply 2.5 million glaciated cells per region. Assuming 10 ice-free grid cells per glaciated cell (i.e., somewhat denser glaciation than the Alps, where we ran with about 20 ice-free cells per glaciated cell), the computational domain would contain 25 million cells, or about 5 times as many as on the 100-m Alps grid. While substantial, the increased computational cost would not be prohibitive.



## 6   Model limitations and future work

### 6.1   SMB scheme



**Figure 10.** (a, d) Difference between 2003 CISM simulated and F19 thickness (m) for select glaciers in the Mont Blanc massif (top row) and the Bernese Alps (bottom row). (b, e) Glacier slope and (c, f) aspect, from the surface elevation DEM.

A limitation in the current modeling framework is the surface mass balance scheme (Sect. 3.2), which is based on a temperature-index approach with a single glacier-specific degree-day factor $\mu$. We adopted this approach for its simplicity in the initial implementation, and also to make best use of the available datasets while accounting for their limitations.

For many glaciers, especially large ones, it is unrealistic to assume that a single degree-day factor applies to the entire glacier. The accumulation/ablation rates depend on terrain characteristics, microclimate effects, debris cover (Rounce et al., 2021), avalanches, and wind drift, among other factors. We refer the reader to Schuster et al. (2023) for a comprehensive





assessment on the effects of SMB schemes, calibration approaches, and parameter values (e.g., lapse rate) on the accuracy of glacier projections.

We illustrate some limitations of the current SMB scheme by focusing on individual glaciers in two regions. The first is the Glacier du Tour, on the northwest side of the Mont Blanc massif. Figure 10a shows the thickness difference between the CISM spin-up and the F19 thickness estimates. This glacier has a tributary that lies between 2800 and 2900 m (dark purple shade in Figure 10a), just above the equilibrium-line altitude (ELA). CISM, however, balances the glacier-wide SMB by choosing a value of $\mu$ that puts the ELA around 2900 m, above the tributary. As a result, this region has a negative SMB, and since it is not fed by any ice in the accumulation zone, it remains ice-free. CISM's SMB scheme cannot account for ELA variations due to aspect and shading. This tributary has a north-facing aspect, ranging from northeast to northwest, and is shaded by steep ridges to the southwest (Figure 10b,c), which could explain why its ELA lies below the glacier mean.

Next, consider the Kander and Tellin Glaciers in the western part of the Bernese Alps (Figure 10d-f). These glaciers are separated by a ridge at around 3000 m elevation, which divides north-flowing glaciers in the Bern canton from south-flowing glaciers in Valais. Both glaciers have regions where the simulated ice is much thinner than the F19 estimates. For Kander, the thin ice along the northwest lateral margin is likely a function of slope. CISM puts ice along glacier walls which in reality are too steep to retain snow and ice.

The Tellin Glacier occupies a narrow elevation range between about 2800 and 3000 m. Since the glacier faces south and southeast (Figure 10f), it receives direct sunlight leading to high melt and a relatively high ELA and terminus. Near the ridge, CISM simulates ice that is much thinner than F19. The reasons for the difference are not obvious, but one possibility is that with the steep slope in the ablation zone, CISM simulates fast flow that moves ice efficiently downslope away from the ridge.

To better capture topographic effects, in future work we plan to introduce new SMB and redistribution schemes within the land component of CESM, the Community Land Model (CLM; Lawrence et al. 2019), which computes surface mass balance using an energy-based scheme. With the hillslope hydrology configuration of CLM (Swenson et al., 2019), we can account for aspect, relief, and slope and assess spatial variability in snow cover, debris cover, and local climate factors. CISM–CLM coupling is currently limited to ice sheets, but we plan to add glacier coupling in future CESM versions, with the aim to improve SMB estimates.

## 6.2 Limitations and uncertainty in atmospheric forcing

High-resolution, high-fidelity land and glacier models often require better-resolved, higher-quality data than their simpler counterparts; otherwise their accuracy is often constrained by the lower resolution of atmospheric and input data. For example, Gabbi et al. (2014) found that a full energy-balance model is less consistent with observations than simpler models, despite being able to capture relevant physical processes in detail. The energy-balance model is more sensitive than simpler models to errors in the atmospheric forcing. Nevertheless, improvements in satellite data and the application of machine learning to data assimilation are gradually resolving these issues.



### 6.3 Missing processes

This initial implementation applies to land-terminating glaciers only, whereas many RGI regions (including Alaska, Arctic Canada, Svalbard, the Russian Arctic, the Greenland periphery, and Antarctic islands) have glaciers that flow into the ocean. In other regions, such as High Mountain Asia and Patagonia, many glaciers terminate in lakes. While CISM includes subgrid parameterizations for basal sliding, iceberg calving, and grounding-line migration (Lipscomb et al., 2019; Leguy et al., 2021), we have not yet implemented these processes for mountain glaciers. This will require a glacier calving parameterization, along with a modified SMB calibration scheme that accounts for the additional mass loss. Finally, CISM currently does not account for glacier surges, which would require a more sophisticated treatment of subglacial hydrology. This is left for future work.

## 7 Conclusions

We have implemented a new framework for modeling mountain glaciers using the Community Ice Sheet Model, which was originally developed to study ice sheets. We used this framework to study the evolution of the nearly 4000 glaciers of the European Alps. This is one of the first uses of a 3D, higher-order ice-flow model to simulate thousands of glaciers at the scale of an RGI region. Unlike traditional flowline models, a 3D higher-order model can resolve complex topography and ice flow patterns, simulating horizontal and vertical variations in velocity, temperature, and stress, thus providing more detailed glacier predictions in complex terrains.

We initialized the model on a 100-meter grid to a stable state using 1980s atmospheric data (monthly average temperature and precipitation), a period when we assume the Alpine glaciers to be near equilibrium with the climate. During the spin-up, we calibrated surface-mass-balance and basal-friction parameters to optimize agreement with observation-based area and thickness targets. We then ran the model forward to the present, simulating glacier retreat in a warming climate. We obtained strong agreement with RGI glacier outlines, consensus ice thickness estimates, and satellite-observed ice velocities.

Using the GlacierMIP3 protocols, we ran the model to equilibrium under various forcing scenarios corresponding to five different CMIP6 Earth system models, several historical and future periods, and three future scenarios. With the present-day climate forcing (2000–2019), we found that the glaciers of the Alps are committed to lose nearly half their area and volume relative to the present day. We simulated near-total ice loss in warmer climate scenarios, with most of the loss taking place before 2100. Climate sensitivity for the Alps in CISM is similar to that of other GlacierMIP3 models analyzed by Zekollari et al. (2024).

We have shown that large-scale, high-resolution, decade-to-millennial-scale glacier simulations can be run at reasonable cost on high-performance computers, due to CISM's parallel scalability, efficient higher-order velocity solver, and methods for limiting the computational domain to active glacier regions. Similar simulations for larger RGI regions should be computationally feasible, and would enable further assessments of the sensitivity of glacier projections to model complexity. Comparing a variety of models with varying complexity will enhance our understanding of glacier systems.

This work suggests many possible areas for improvement: for example, (1) using the Community Land Model within the CESM framework to improve SMB calculations by adding physical factors and terrain characteristics such as aspect, slope,



and debris cover, (2) extending the model to lake/marine-terminating glaciers, which would require a treatment of calving, and (3) modeling glacier surges, which would demand more sophisticated subglacial hydrology.

500      Overall, these simulations are a step towards unification of glacier and ice sheet studies in a common modeling framework. Further, this model development will enable assessments on the interactions and feedback mechanisms between land ice and interconnected processes including freshwater resources, ecosystem dynamics, and sea-level rise using the Community Earth System Model.

*Code and data availability.* CISM is an open-source code developed on the Earth System Community Model Portal (ESCOMP) Git repos-
505  itory at https://github.com/ESCOMP/CISM. The version used to perform the glacier simulations is the CISM release version 2.2, tagged as cism_main_2.02.001 on the GitHub repository (https://github.com/ESCOMP/CISM/releases/tag/cism_main_2.02.001; last access: 24 January 2025). Zenodo also hosts the archived model source code (v2.2), along with the configuration files and input data (10.5281/zenodo.14714941; Minallah et al. 2025), and the resulting model output for GlacierMIP3 (10.5281/zenodo.14045268; Schuster et al. 2024).

*Author contributions.* SM and WHL designed the study and the modeling framework. SM compiled input datasets and led the validation,
510  analysis, and output assessments. WHL implemented the new glacier module and related software infrastructure in CISM. GL provided support for data processing and ran the CISM–GlacierMIP3 simulations. HZ assisted in implementing the GlacierMIP3 protocols and provided guidance on the current glacier modeling frameworks in literature. SM and WHL co-wrote the manuscript, with SM creating the figures. All authors discussed the modeling framework and the final manuscript.

*Competing interests.* The authors declare no competing interests.

515  *Acknowledgements.* This material is based upon work supported by the NSF National Center for Atmospheric Research (NCAR), which is a major facility sponsored by the U.S. National Science Foundation under Cooperative Agreement No. 1852977. We acknowledge high-performance computing support from the Derecho system (doi:10.5065/qx9a-pg09) and computing resources provided by the Climate Simulation Laboratory at NSF NCAR's Computational and Information Systems Laboratory (CISL). SM was additionally funded by the NSF NCAR Advanced Study Program (ASP) postdoctoral fellowship. HZ was funded by the European Research Council (ERC) under the Euro-
520  pean Union's Horizon Framework research and innovation programme, grant agreement 101115565 and the Research foundation – Flanders (FWO) through an Odysseus Type II project, grant agreement G0DCA23N.

     We thank Fabien Maussion and Lilian Schuster for their support and guidance, especially in designing the surface-mass-balance scheme in CISM. We also thank Kate Thayer-Calder for CISM software engineering support, including the transition to the NSF NCAR Derecho supercomputer.



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
