# Peer review of "A framework for three-dimensional dynamic modeling of mountain glaciers in the Community Ice Sheet Model (CISM v2.2)"

_EGUsphere, 2024_

## Referee Comment (RC1)

**Review of Minallah et al. (2025) 'A framework for three-dimensional dynamic modeling of mountain glaciers in the Community Ice Sheet Model (CISM v2.2)'**

**Summary**
This paper details a set of related modelling developments that allow CISM to be applied to regional studies of mountain glaciers, rather than its native ice-sheet domains. The authors develop a protocol to allow the glaciers to be initialised in a manner similar to how an ice sheet would be, and also solve the computational-time problem by having ice-free blocks of the model domain ignored during simulations (these blocks update dynamically if glaciers advance/retreat). CISM is then applied to the European Alps using the full suite of GlacierMIP3 experiments to demonstrate the effectiveness of this new approach, with results showing that it performs in line with expectations based on previous studies.

I think this paper is well-written and put-together. I would particularly like to congratulate the authors on their concluding sections on model limitations and future work, which addressed several queries I'd had at the back of my mind while reading the rest of the manuscript, and allowed me to write a much shorter review! The technical advance presented in the paper is novel, particularly the ability to ignore ice-free blocks of the model domain, and the possibility it raises of having a unified ice-sheet-glacier representation in the same model is, to my mind, the major advantage it presents (whilst this is technically possible in other models, it is not often done, certainly not in a sustainable way). It is also clear that the model functions well.

My comments below are largely fairly minor and most should be able to be addressed with a line or two of extra explanation. There is some amount of restructuring, though, that I think would improve the paper, but, overall, I think this fits within the broader category of minor revisions.

Page and line numbers refer to those in the clean version of the submitted manuscript.

**Major Comments**

- Spin-up: This is not a problem *per se*, but I think the 10,000-year spin-up for the glaciers seems a little excessive. Clearly, too much spin-up is far better than too little, but the paper does not justify why such a long period is necessary for such small ice bodies. Especially given the authors note in Section 4.6 that the glaciers are close to equilibration with the warmer 2010 temperature conditions by 2184 (so, 174 years). Given the authors are not starting from ice-free topography or similar, but with an initial glacier that is not all that far from the stable 1984 state they are aiming for, I can't believe that it takes 10,000 years to sort itself out. I suspect the authors could drop an order of magnitude and have 800 years of inversion for $C_p$ followed by 200 years of holding it constant without the results being materially different, and even this might be at the upper end of what I'd think is reasonable in this setting. The main weakness of the method is that it does need a substantial amount of HPC resource to run and there seems to be a very low-hanging fruit here that could greatly improve the situation. Could the authors at least provide some commentary in the paper on why this length of spin-up is necessary or, if it isn't, note that this is the case (I'm not expecting the authors to check whether a shorter spin-up does affect the results, given the computational cost, but I would like some explanation of the reasoning here)?
- Structure: The paper is generally well-structured, but I found the presentation of the method in a theoretical manner in Section 3, followed by the concrete application details in Section 4 a bit confusing. I read Section 3 expecting to find a lot of details that come up later and then had to keep cross-referencing the theory and the application, which was a bit annoying (for instance, I read Section 3.3 and was very concerned that the authors hadn't said anything about the datasets they were using, and then had to wait till Section 4.2 to find out what had actually happened, then check back to 3.3 to remind myself what the actual method was. This seems as if it could be simpler). I might recommend that the authors consider merging the two sections and restructuring such that each bit of theory is followed by how it was implemented in this study, just to make it easier to follow what's going on

**Minor Comments**

- p.4, l.91: Could the authors expand a little here and say in which glaciated regions DIVA does not tend to match Blatter-Pattyn? It will make it easier for readers to understand where this model formulation is likely to be more/less reliable.
- p.11, l.286: I think these spin-up times could have a 0 knocked off them and the results would not change very much at all. Out of interest, why did the authors choose such long simulations?
- p.14, l. 309-315: I agree that it's good to compare to the Farinotti product, but there's no particular reason to assume that it is a completely accurate reflection of reality. In fact, we know it isn't. My point is that deviations between the results presented here and the Farinotti product are not necessarily a bad thing. It would be instructive to also compare the thickness results to the products in Millan et al. (2022) and Cook et al. (2023) to see where the results here fall within the range of existing global-regional modelled Alpine thickness products, rather than just picking one and assuming it's the best representation. All three products work better in some places and worse in others, so a wider comparison might be more useful for the community to understand how the method presented here performs and what it offers that isn't already on the table. I would also make a similar point about Section 6.1 – the differences may be more or less marked if other products are also considered, which might say something useful about the method presented here
- p.21, l.395: OK, both resolutions produce stable runs. Are there any significant differences in the actual results? That seems a critical point that the authors should address in this section (I assume the differences were pretty minor, but it should be clearly stated here, given the section title!)

**References**

Cook, Samuel J., Guillaume Jouvet, Romain Millan, Antoine Rabatel, Harry Zekollari, and Inés Dussaillant. "Committed Ice Loss in the European Alps Until 2050 Using a Deep-Learning-Aided 3D Ice-Flow Model With Data Assimilation." *Geophysical Research Letters* 50, no. 23 (2023): e2023GL105029. https://doi.org/10.1029/2023GL105029.

Millan, Romain, Jérémie Mouginot, Antoine Rabatel, and Mathieu Morlighem. "Ice Velocity and Thickness of the World's Glaciers." *Nature Geoscience* 15 (February 7, 2022): 124–29. https://doi.org/10.1038/s41561-021-00885-z.

---

## Author Comment (AC1)

**Author's response to RC1**

*Text in black: Reviewer's comments*
*Text in blue: Author's response*

**Summary**

This paper details a set of related modelling developments that allow CISM to be applied to regional studies of mountain glaciers, rather than its native ice-sheet domains. The authors develop a protocol to allow the glaciers to be initialised in a manner similar to how an ice sheet would be, and also solve the computational- time problem by having ice-free blocks of the model domain ignored during simulations (these blocks update dynamically if glaciers advance/retreat). CISM is then applied to the European Alps using the full suite of GlacierMIP3 experiments to demonstrate the effectiveness of this new approach, with results showing that it performs in line with expectations based on previous studies.

I think this paper is well-written and put-together. I would particularly like to congratulate the authors on their concluding sections on model limitations and future work, which addressed several queries I'd had at the back of my mind while reading the rest of the manuscript, and allowed me to write a much shorter review! The technical advance presented in the paper is novel, particularly the ability to ignore ice-free blocks of the model domain, and the possibility it raises of having a unified ice-sheet-glacier representation in the same model is, to my mind, the major advantage it presents (whilst this is technically possible in other models, it is not often done, certainly not in a sustainable way). It is also clear that the model functions well.

My comments below are largely fairly minor and most should be able to be addressed with a line or two of extra explanation. There is some amount of restructuring, though, that I think would improve the paper, but, overall, I think this fits within the broader category of minor revisions.

We thank the reviewer for their very constructive and encouraging feedback. We have incorporated their suggestions and provided detailed responses below.

Page and line numbers refer to those in the clean version of the submitted manuscript.

**Major Comments**

- Spin-up: This is not a problem *per se*, but I think the 10,000-year spin-up for the glaciers seems a little excessive. Clearly, too much spin-up is far better than too little, but the paper does not justify why such a long period is necessary for such small ice bodies. Especially given the authors note in Section 4.6 that the glaciers are close to equilibration with the warmer 2010 temperature conditions by 2184 (so, 174 years). Given the authors are not starting from ice-free topography or similar, but with an initial glacier that is not all that far from the stable 1984 state they are aiming for, I can't believe that it takes 10,000 years to sort itself out. I suspect the authors could drop an order of magnitude and have 800 years of inversion for $C_p$ followed by 200 years of holding it constant without the results being materially different, and even this might be at the upper end of what I'd think is reasonable in

this setting. The main weakness of the method is that it does need a substantial amount of HPC resource to run and there seems to be a very low-hanging fruit here that could greatly improve the situation. Could the authors at least provide some commentary in the paper on why this length of spin-up is necessary or, if it isn't, note that this is the case (I'm not expecting the authors to check whether a shorter spin-up does affect the results, given the computational cost, but I would like some explanation of the reasoning here)?

During the forward runs, the glaciers adjust to the warmer climates relatively quickly (~200 years). During the model initialization, however, it takes longer for the simulated ice geometry and internal state to reach quasi-equilibrium with the climate forcing. Below, we show the total glacier volume for the Alps during the original 10,000-year spin-up (x-axis). After a large response in the first few centuries, the volume slowly equilibrates over a few thousand years.

[Figure]

We recognize, however, that changes are very small after the first 4,000 years, and we do not want to give readers the impression that a full 10,000-year spin-up is necessary.

For the revised submission, we ran a new spin-up of 5,000 years (4,000 years with inversion and 1,000 years without inversion). We stopped the inversion once the rate of change of total ice volume fell below $0.1$ km$^3$ per 1,000 years.

We describe the modified spin-up and equilibrium criterion in Section 4.1 (*Spin-up and historical runs*) in the revised manuscript.

- Structure: The paper is generally well-structured, but I found the presentation of the method in a theoretical manner in Section 3, followed by the concrete application details in Section 4 a bit confusing. I read Section 3 expecting to find a lot of details that come up later and then had to keep cross-referencing the theory and the application, which was a bit annoying (for instance, I read Section 3.3 and was very concerned that the authors hadn't said anything about the datasets they were using, and then had to wait till Section 4.2 to find out what had actually happened, then check back to 3.3 to remind myself what the actual method was. This seems as if it could be simpler). I might recommend that the authors consider merging the two sections and restructuring such that each bit of theory is followed by how it was implemented in this study, just to make it easier to follow what's going on

Thank you for the feedback! We have revised the structure of the manuscript to merge the

two sections. The revised version combines methodological theory and application within the same sections to improve clarity and flow.

The structure for the new Sections 3 - 5 is as follows:

3. CISM development for glacier modeling and application to the European Alps
       3.1 Forcing and initialization data
       3.2 Model domain and resolution
       3.3 Glacier identification and tracking
       3.4 Glacier surface mass balance scheme

4. Model initialization and calibration
       4.1 Spin-up and historical runs
       4.2 Thickness inversion
       4.3 Mass balance calibration
       4.4 Surface velocity

5. GlacierMIP3 simulations
       5.1 Committed ice loss
       5.2 Equilibration runs

**Minor Comments**
- p.4, l.91: Could the authors expand a little here and say in which glaciated regions DIVA does not tend to match Blatter-Pattyn? It will make it easier for readers to understand where this model formulation is likely to be more/less reliable.

DIVA differs from Blatter-Pattyn (BP) for flow with large vertical shear over a bed with significant topographic variations on short spatial scales, as discussed by Goldberg (2011). A classic example is Test A in the higher-order benchmark experiments introduced by Pattyn et al. (2008). In our experiments, the flow of many glaciers is dominated by sliding. For a few large glaciers, e.g., the Aletsch in the Bernese Alps, there is significant vertical shear, but the bed topography is fairly smooth, so the differences between BP and DIVA are relatively small.

We clarified these points with new text in the revised Section 2.1:

*"The DIVA solver is computationally much faster than BP, while computing velocities similar to BP in most glaciated regions. An exception would be flow with large vertical shear over a bed with rough topography, as discussed by Goldberg (2011). In the runs below, most glaciers have relatively smooth beds and/or sliding-dominated flow with small vertical shear, for which the two solvers give similar results. DIVA also scales well to the high resolutions needed to model mountain glaciers (Robinson et al., 2022). We therefore used DIVA for the simulations in this study."*

To illustrate this, we ran a 1,000-year BP simulation on the 200-m grid and compared it with the same DIVA set-up. Since BP is an order of magnitude more expensive than DIVA, we did not carry out a full spin-up with BP.

The figure below shows ice thickness and velocity differences for DIVA versus BP in the

Bernese Alps. The plot shows small differences compared to the mean values, especially in the thickness field.

[Figure]

The comparison of the two schemes for the Alps glaciers is not included in the revision as we deemed it outside the paper's scope.

- p.11, l.286: I think these spin-up times could have a 0 knocked off them and the results would not change very much at all. Out of interest, why did the authors choose such long simulations?

For the revised submission, we reduced the spin-up time to 5,000 years. It is true that the results would be fairly similar had we stopped at year 1000, but we took a conservative approach and continued the spin-up until the rate of change of total volume fell below 0.1 km$^3$ per 1000 years.

- p.14, l. 309-315: I agree that it's good to compare to the Farinotti product, but there's no particular reason to assume that it is a completely accurate reflection of reality. In fact, we know it isn't. My point is that deviations between the results presented here and the Farinotti product are not necessarily a bad thing. It would be instructive to also compare the thickness results to the products in Millan et al. (2022) and Cook et al. (2023) to see where the results here fall within the range of existing global-regional modelled Alpine thickness products, rather than just picking one and assuming it's the best representation. All three products work better in some places and worse in others, so a wider comparison might be more useful for the

community to understand how the method presented here performs and what it offers that isn't already on the table. I would also make a similar point about Section 6.1 – the differences may be more or less marked if other products are also considered, which might say something useful about the method presented here

We used the Farinotti product (F19) as the target thickness for calibration (Section 3.1), and these results (Figure 2 and Figure 10 in the revised manuscript) are discussing how well CISM reaches this target thickness.

We agree that the F19 data is far from accurate. To ensure that our model simulations are within the range of existing estimates, we compared our output with the Grab et al. thickness data (Figure 3) and with the ITS-LIVE and Millan et al. 2022 velocity profiles in Figure 5 (instead of their thickness, which was a derived product).

- p.21, l.395: OK, both resolutions produce stable runs. Are there any significant differences in the actual results? That seems a critical point that the authors should address in this section (I assume the differences were pretty minor, but it should be clearly stated here, given the section title!)

We agree that it is useful to compare results on both 100- and 200-m grids. For the revised submission, Section 3.2 (*Model domain and resolution*) and Section 6.1 (*Spatial resolution and computational efficiency*) provides further comparison and results for a spin-up and forward runs between the two grids, with the same model settings apart from resolution.

Indeed, the differences between the two runs are small. The 200-m run has a slightly larger total ice area and volume at the end of the spin-up (2530.8 km$^2$ and 137.9 km$^3$, respectively, compared to 2497.2 km$^2$ and 137.3 km$^3$ for the 100-m run), however its sensitivity to warming is very similar.

The main advantage of the 100-m grid is that we are able to resolve more of the small glaciers. The 200-m grid has a much larger number of unresolved (subgrid) glaciers and glaciers occupying just one or two grid cells (table below).

| Total glaciers in RGI: 3892 | | |
|---|---|---|
| | 100-m | 200-m |
| Unresolved | 6 | 259 |
| 1 grid | 42 | 782 |
| 2 grid | 182 | 493 |

**References**

Cook, Samuel J., Guillaume Jouvet, Romain Millan, Antoine Rabatel, Harry Zekollari, and Inés Dussaillant. "Committed Ice Loss in the European Alps Until 2050 Using a Deep-Learning-Aided 3D Ice-Flow Model With Data Assimilation." *Geophysical Research Letters* 50, no. 23 (2023): e2023GL105029. https://doi.org/10.1029/2023GL105029.

Goldberg, D. N.: A variationally derived, depth-integrated approximation to a higher-order glaciological flow model, J. Glaciol., 57, 157–170, https://doi.org/10.3189/002214311795306763, 2011.

Millan, Romain, Jérémie Mouginot, Antoine Rabatel, and Mathieu Morlighem. "Ice Velocity and Thickness of the World's Glaciers." *Nature Geoscience* 15 (February 7, 2022): 124–29. https://doi.org/10.1038/s41561- 021-00885-z

Pattyn, F., Perichon, L., Aschwanden, A., Breuer, B., de Smedt, B., Gagliardini, O., Gudmundsson, G. H., Hindmarsh, R. C. A., Hubbard, A., Johnson, J. V., Kleiner, T., Konovalov, Y., Martin, C., Payne, A. J., Pollard, D., Price, S., Rückamp, M., Saito, F., Souˇcek, O., Sugiyama, S., and Zwinger, T.: Benchmark experiments for higher-order and full-Stokes ice sheet models (ISMIP—HOM), The Cryosphere, 2, 95–108, https://doi.org/10.5194/tc-2-95-2008, 200

---

## Author Comment (AC2)

**Author's response to RC2**

*Text in black: Reviewer's comments*
*Text in blue: Author's response*

This manuscript introduces a 3D dynamics model for mountain glaciers based on the Community Ice Sheet Model, and its application to study the evolution of mountain glaciers using protocols from the third phase of the Glacier Model Intercomparison Project (GlacierMIP3). While the paper is overall well written, I have several minor concerns that should be addressed before the manuscript is considered suitable for publication.

We thank the reviewer for their constructive feedback. We have incorporated their suggestions and provide detailed responses to their comments below.

I would like the authors to better motivate, in the Introduction, the need to use high-fidelity models for mountain glaciers, particularly considering their increased sensitivity to data errors. The authors touch on this in the Discussion, but this should be featured in the Introduction as well.

We have included a motivation to include glacier modelling within an ESM framework in the introduction:

*"Integrating glacier modeling within an Earth System Model (ESM) framework offers several emerging advantages for studying glaciated regions. It will enable dynamic coupling with the climate system, enhancing the representation of feedback mechanisms with the land-atmosphere-hydrology components. This integration will allow more comprehensive assessments of climate, ecological, and hydrological impacts across glaciated regions worldwide."*

The initialization of the model is done with ad hoc tuning methods (e.g., Pollard and DeConto, 2012). However, I think the reader would benefit from a discussion of more advanced initialization techniques based on (transient) PDE-constrained optimization methods, which have become increasingly standard in the literature and offer a more rigorous approach to model initialization.
Pollard, D. and DeConto, R. M.: A simple inverse method for the distribution of basal sliding coefficients under ice sheets, applied to Antarctica, The Cryosphere, 6, 953–971, https://doi.org/10.5194/tc-6-953-2012, 2012.

We agree with the reviewer on the use of more sophisticated approaches in literature for initialization. We used the approach described in the manuscript for its simplicity, computational efficiency, and ease of implementation, and because we have used a similar method for ice sheet studies.

To address the reviewer's concern, we have included a new Section 7.2 on "*Glacier initialization*" under Section on "Model limitations and future work".

*"CISM uses a simple, computationally efficient inverse method to estimate the spatial distribution of the basal friction coefficient Cp, adjusting the values to minimize the mismatch between modeled and observed ice thickness (Sect. 4.2). It is similar to the method developed by Pollard and DeConto (2012) to derive basal sliding coefficients for Antarctica. Other*

*studies have used more sophisticated approaches such as adjoint-based optimization methods, which compute gradients of a cost function (typically the mismatch between modeled and observed velocities) with respect to control parameters using the adjoint of the governing equations, or transient (time-evolving) inversion that assimilates time series of observations (Morlighem et al., 2013; Goldberg and Heimbach, 2013; Perego et al., 2014). At present, CISM does not have these capabilities for ice-flow modeling."*

**Detailed comments:**

Equation (1): For clarity and completeness, please include the y-component of the DIVA model alongside the x-component.

Done - the equation for the y-direction is also added:

$$\frac{1}{H}\frac{\partial}{\partial x}\left[2\bar{\eta}H\left(2\frac{\partial \bar{u}}{\partial x}+\frac{\partial \bar{v}}{\partial y}\right)\right]+\frac{1}{H}\frac{\partial}{\partial y}\left[\bar{\eta}H\left(\frac{\partial \bar{u}}{\partial y}+\frac{\partial \bar{v}}{\partial x}\right)\right]+\frac{\partial}{\partial z}\left(\eta\frac{\partial u}{\partial z}\right)=\rho_i g\frac{\partial s}{\partial x},$$
$$\frac{1}{H}\frac{\partial}{\partial x}\left[\bar{\eta}H\left(\frac{\partial \bar{u}}{\partial y}+\frac{\partial \bar{v}}{\partial x}\right)\right]+\frac{1}{H}\frac{\partial}{\partial y}\left[2\bar{\eta}H\left(\frac{\partial \bar{u}}{\partial x}+2\frac{\partial \bar{v}}{\partial y}\right)\right]+\frac{\partial}{\partial z}\left(\eta\frac{\partial v}{\partial z}\right)=\rho_i g\frac{\partial s}{\partial y},$$

(1)

Equation (2): Please clarify whether the cap applies to the absolute values of the separate x and y derivatives of the surface elevation, or to the magnitude of the surface elevation gradient. Either way, this seems a bit of a crude fix. How frequently is this correction active in simulations performed in this study?

The cap applies to the absolute values of the separate x and y derivatives at each cell edge. We have added the clarification in the revised text (Section 2.1):

*"When solving Eq. (1), CISM has an option to cap the magnitude of the surface slope ($\|\partial s/\partial x\|$ at east and west cell edges and $\|\partial s/\partial y\|$ at north and south edges) at a value of $m_{max}$ to maintain model stability in regions of steep topography. For this study we set $m_{max} = 1.0$ (a 45° slope). The cap is applied to about 1% of ice-covered cell edges in the Alps domain described in (Sect. 4)."*

We agree that slope-limiting is a crude fix. We have used this approach in Greenland and Antarctic ice-sheet simulations where relatively few cells exceed the limit, but in the steep terrain of the Alps, the limit was being applied to a substantial fraction of the ice-covered cells.

For the revised simulation, we carried out a new set of runs with $m_{max} = 1.0$, i.e. an angle of 45° relative to the horizontal. With this change, the slope is limited at only about 1% of cell edges. When we first made this change, flow speeds increased over steep terrain and the spun-up ice was too thin. However, we found that increasing the values of the $C_p^{min}$, $C_p^{init}$, and $C_p^{max}$ parameters from 3K/30K/100K to 5K/50K/200K reduced the maximum speeds (so that we did not have to reduce the 1-month timestep) and enabled a good match to the target thicknesses. The rms error between the simulated and target thickness is now 12.5 m, compared to 14.8 m in the previous submission. This suggests that minimizing the use of the slope cap has improved the representation of the flow.

We highlight that the new results, based on these changes, are consistent with the original simulations, and our analysis remains unchanged.

Equation (3): The scalar representation of the sliding law is potentially confusing, as basal shear stress and basal velocity are vector fields.
Please rewrite this equation in vector form, for example: tau_b = C_p |u_b|^{1/m-1} u_b.

Done - we have modified the equation as suggested:

$$\tau_b = C_p |\mathbf{u}_b|^{\frac{1}{m}-1} \mathbf{u}_b,$$

Section 5.1: Data Blocks and Repartitioning: The term "data block" needs to be defined. Are data blocks the fundamental units used for distributing computational workload across processor cores? How is the repartitioning of the data blocks performed, especially considering they are now unstructured?

Yes, the blocks are the fundamental units for distributing the workload. Each square block of data is assigned to one processor core. The resulting blocks are not unstructured. Rather, we can think of them as squares on a checkerboard in which some squares are labeled as active, while the rest are labeled as inactive. The active blocks are initialized with information that tells them which adjacent blocks are also active. During the run, each active block exchanges information only with neighboring blocks that are also active.

We have added additional information in Section 6.1 (*Spatial resolution and computational efficiency*) to clarify how this works:

*"On initialization, CISM partitions the global domain into square blocks of data. For the 100-m grid, the initial domain consists of about 9300 blocks, each with 75 or 76~cells on a side. Of these blocks, only about 900 contain one or more of the grid cells included in the glacier mask. CISM labels these blocks as active, assigns one block to each processor core, and discards the remaining blocks. We modified CISM's parallel routines (halo updates, gather/scatters, global sums, and broadcasts) to operate only on active blocks, exchanging data with adjacent blocks that are also active. This allows a tenfold reduction in cost compared to a simulation with inactive blocks included."*